# Fungal Colonization and Infections—Interactions with Other Human Diseases

**DOI:** 10.3390/pathogens11020212

**Published:** 2022-02-06

**Authors:** Shanmuga S. Mahalingam, Sangeetha Jayaraman, Pushpa Pandiyan

**Affiliations:** 1Department of Biological Sciences, School of Dental Medicine, Case Western Reserve University, Cleveland, OH 44106, USA; sxm1505@case.edu (S.S.M.); sxj579@case.edu (S.J.); 2Department of Pathology, School of Medicine, Case Western Reserve University, Cleveland, OH 44106, USA; 3Case Comprehensive Cancer Center, Case Western Reserve University, University Hospitals Cleveland Medical Center, Cleveland, OH 44106, USA

**Keywords:** *Candida*, fungi, inflammation, T_reg_, COVID-19, SARS-CoV-2, mycobiome

## Abstract

*Candida albicans* is a commensal fungus that asymptomatically colonizes the skin and mucosa of 60% of healthy individuals. Breaches in the cutaneous and mucosal barriers trigger candidiasis that ranges from asymptomatic candidemia and mucosal infections to fulminant sepsis with 70% mortality rates. Fungi influence at least several diseases, in part by mechanisms such as the production of pro-carcinogenic agents, molecular mimicking, and triggering of the inflammation cascade. These processes impact the interactions among human pathogenic and resident fungi, the bacteriome in various organs/tissues, and the host immune system, dictating the outcomes of invasive infections, metabolic diseases, and cancer. Although mechanistic investigations are at stages of infancy, recent studies have advanced our understanding of host–fungal interactions, their role in immune homeostasis, and their associated pathologies. This review summarizes the role of *C. albicans* and other opportunistic fungi, specifically their association with various diseases, providing a glimpse at the recent developments and our current knowledge in the context of inflammatory-bowel disease (IBD), cancers, and COVID-19. Two of the most common human diseases where fungal interactions have been previously well-studied are cancer and IBD. Here we also discuss the emerging role of fungi in the ongoing and evolving pandemic of COVID-19, as it is relevant to current health affairs.

## 1. Introduction

The past century has witnessed a significant increase in the wealth of knowledge on infectious diseases caused by viruses, bacteria, fungi, protozoa, and prions. Among the public health issues caused by microbes, fungal diseases are relatively neglected, owing to the low mortality rate of 1.5% globally [1]. Fungal diseases, also known as mycoses, although acknowledged in the late 1980s, have now captured greater attention than ever before because of the escalating population of high-risk elderly individuals, immunosuppressed individuals, transplant recipients, and premature neonates [2,3,4,5,6,7]. The average life expectancy has increased tremendously with our current scientific knowledge and advancements in medical treatments increasing the population size of individuals over 65 years of age. The United States census reports 39.6 million people aged 65 and over, which is expected to double by 2050, contributing to 21% of the population [8,9]. This population and other immunosuppressed individuals show a significant increase in the prevalence of fungal infections because of environmental factors as well as intrinsic degenerative immune and metabolic changes [10,11,12]. 

Among fungi, the *Candida* species have co-existed as the most common and innocuous commensals associated with human beings for quite a long time. They are frequently encountered on human skin, the mucosal surfaces of gastrointestinal and genitourinary tracts. Nonetheless, in immunologically weak, premature neonates, elderly people, and immunocompromised individuals they become opportunistic pathogens. Pathogenic adaptations by *Candida* manifest as local mucosal infections or systemic infections, occasionally resulting in infections spreading to major organs [13,14]. It ranks fourth as a causal factor among healthcare-associated infections and second among healthcare-related bloodstream infections in the United States [15,16,17,18]. 

Of the several known species of *Candida*, fifteen species distinctively cause human diseases. A majority of infections are caused by five different pathogenic strains, namely *C. albicans*, *C. tropicalis*, *C. krusei* (now known as *Pichia kudriavzevii*), *C. glabrata*, *C. parapsilosis*, and a more recently recognized emerging major pathogen, *C. auris*. The SENTRY Antimicrobial Surveillance Program study that was conducted for 20 years has shown that *C. albicans* represents almost half of the infectious *Candida* cases (47%) worldwide [19]. Furthermore, the same study has revealed a minor increase in non-albicans *C. glabrata* (19%), *C. parapsilosis* (16%), *C. tropicalis* (10%), *C. krusei* (3%), and 6.5% miscellaneous *Candida* species [19]. 

The first point of contact for invading pathogens is the epithelial barrier lining the skin, and the mucosa, followed by the cells of the host’s innate immune system. Antigen-presenting cells, namely dendritic cells (DCs) and macrophages are decisively positioned at the frontline of defense against these intruders. These cells essentially express the germline-encoded pattern-recognition receptors (PRRs) through which they sense several conserved architectures of the fungal cell wall and elicit innate immune responses [20]. These receptors including toll-like receptor-2, which is expressed on a wide variety of immune cells, have been extensively studied over the last two decades for their role in anti-fungal immunity. They sense fungal cell wall components including mannans, mannoproteins, β-glucans, and chitin [20,21,22]. C-type lectin receptors (CLRs) comprising Dendritic cell-associated C-type lectin-1 (Dectin-1), the Dectin-2 cluster (Dectin-2, Mcl and Mincle), Dectin-3, mannose receptor, and DC-specific ICAM-3-grabbing non-integrin (DC-SIGN) also recognize fungal derivatives and are mainly expressed on myeloid cells. Collectins are comprised of mannose-binding lectin and surfactant proteins A and B, and are additional players involved in anti-fungal immunity [23,24,25]. Upon ligation, PRRs induce multiple signaling cascades leading to the production of pro-inflammatory cytokines and antimicrobial proteins by resident macrophages, PMNs and epithelial cells [26]. Dectin-1 senses β-glucans present on the surface of fungi, and Dectin-2 and Dectin-3 sense α-mannans of fungal hyphae [24,25]. Upon the binding of Dectin-1, fungal β-glucans lead to the phosphorylation of the cytoplasmic domain of Dectin-1 by Src kinase and the docking and activation of spleen tyrosine kinase (Syk). These events induce the assembly of a scaffold comprising of caspase-recruitment domain 9 (CARD9), adaptor-protein B-cell lymophoma-10 (Bcl-10) and mucosa-associated lymphoid-tissue-lymphoma-translocation protein 1 (MALT1) [27,28,29]. The assembly and activation of the CARD9/Bcl-10/MALT1 complex activates the canonical NF-κB pathway leading to the release of various pro-inflammatory cytokines and reactive oxygen species (ROS) production, thereby triggering phagocytosis (Figure 1) [30,31]. 

Besides direct infections and morbidities, there is an accumulative concern about *Candida* colonization and invasion associated with several pathophysiological disorders including inflammatory-bowel disease (IBD), cancer, diabetes, metabolic diseases, and other infectious and inflammatory conditions. Notably, all of these are closely linked with underlying dysbiosis, i.e., changes in the resident microbiome. Although the initial research from 1998 focused solely on bacteriome appraising the bacterial community [32,33], some of the recent investigations have more comprehensively focused on characterization and exploration of the fungal biota. Thus, the fungal microbiome, namely the mycobiome, is an emerging integral component of the human microbiome but its precise function remains understudied [34,35,36]. *Candida,* being the highly represented genera in the mycobiome, is best-studied for its relationship with the host as an opportunistic pathogen as well as its non-pathogenic interactions and contributions to the host’s immune development [37]. The invasion by this yeast occurs through its morphological transition to the hyphal form thereby breaching mucosal barriers and gaining access to underlying host tissues (Figure 1) [38,39]. The mechanisms of barrier breach and the innate factors produced by the host are reviewed elsewhere [17,21,40,41,42]. This review sheds light on recent studies on how fungal components including those from *Candida* are mechanistically involved in human health and the diseases discussed here.

## 2. Association of *Candida* in Development and Progression of Cancer

Cancer is among the most prominent causes of death worldwide, posing an important threat to life expectancy in every country regardless of the level of economic development [43]. Based on the recent study by GLOBOCAN 2020, an estimate of 20 million new cancer cases have been reported and ~10 million deaths have occurred in 2020 alone [44]. Lately, the relationship between cancer and microbial infections has attracted enormous attention [45]. Transient immunosuppression is commonly seen in cancer patients undergoing chemotherapy, which consequently has been linked to bacterial and fungal infections. Recent studies provide insights on the association between the presence of microorganisms and the augmented risk of cancer development. The role of microbes and their involvement in diverse mechanisms including the initiation, establishment, and spread of cancer is reviewed elsewhere [46,47]. Here we will review in particular the contribution of colonization and infections by *Candida* species and their role in cancer. 

Although *Candida* species are capable of fueling the initiation and propagation of cancerous processes, they are not by themselves oncogenic [48,49,50,51,52,53]. Candidiasis, although a predictor of cancer risk, may also result from cancer and their establishment could be favored by the immunosuppression resulting from cancer chemotherapy. Fungal infections modulate cancer development by impacting the host through diverse mechanisms: (1) perturbations in the DNA-damage response cause genetic mutations that accumulate inside the cell, modifying the oncogene expression involved in cell survival and proliferation; (2) oncogenic inflammation induced by DNA-damaging fungal toxins and their carcinogenic-inducing metabolites; and (3) fungal colonization or infection resulting in intense inflammation favoring the growth of primary tumors and metastases, making tumors resistant to chemotherapy drugs and suppressing the host’s anti-cancer immune responses [45]. 

In normal healthy individuals, *Candida* species are commonly found as a commensal colonizer in the skin and mucous membranes including the nose, mouth, gastrointestinal tract, reproductive organs, etc. [41,54]. Understandably, *Candida* burden has been shown to positively correlate with the progression of oral, esophageal, and colorectal cancer, which is consistent with their presence in these mucosa [55,56,57]. In a recent study of 100 patients inflicted with oral squamous-cell carcinoma (OSCC), 75% of the patients had *Candida* species [58]: 84% of those identified were *C. albicans*, and other non-albicans species varied between 1% and 8%. Patients with hematopoietic neoplasms, head and neck malignancies as well patients undergoing chemo/radiotherapy have also presented with oral candidiasis ranging from 7% to 52% [59,60]. Studies on animal models have also indicated that infections caused by *C. albicans* can contribute to carcinogenesis similar to known carcinogenic substances [45,48,55]. Several studies have shown substantial associations between candidiasis and dysplasia in the oral cavity, precancerous disorders, and OSCC [56,61,62].

A widely regarded hypothesis for the pro-carcinogenic effect of *Candida* species is the direct production of carcinogens and/or the metabolism of pro-carcinogens among other molecular mechanisms. Studies using Sprague–Dawley rats showed that nitrosamine production and release occurring through the hyphal invasion caused by oral microbiota dysbiosis leads to oral tumor growth and progression [63,64]. These findings are in line with the initial studies establishing that *C. albicans* can act as a promoter of carcinogenesis in the rat/mouse tongue following recurrent applications of nitroquinoline (4-nitroquinoline 1-oxide; 4-NQO), thereby mimicking the human neck and head cancer [65,66]. Furthermore, overexpression of Ki-67, P53, and COX-2 by the host cells following *Candida* infection appears to suggest the involvement of this fungus in the malignant transformation of the host cells [67]. Cell-proliferation markers (Ki-67 and P53) and their overexpression are well-described in several malignancies. Inflammatory marker COX-2, an enzyme that converts arachidonic acid to prostanoids (prostaglandins, thromboxanes and prostacyclins) are expressed in several cancers and precancerous lesions, thereby prompting cell proliferation, tumor invasion, and cell death [55]. 

Oral carcinogenesis begins as an epithelial dysplasia where the normal structure of epithelial cells is altered to an unusual proliferative state. Dysplasia usually results from a cell injury followed by chronic inflammation [68,69]. The characteristic feature of epithelial dysplasia is the altered proliferation of the damaged squamous cells on the epithelial surface leading to degradation of the basal membrane. These damaged cells undergo apoptosis or may transform into a malignant state, thereby resulting in local destruction and distant invasion [70]. 

The excessive production of pro-inflammatory cytokines during oral *Candida* infection such as IL-1α, IL-1β, IL-6, IL-8, TNF-α, IFN-γ, etc. suggests that alterations in metabolic pathways and dysfunction of the endothelium may negatively affect immune-related mechanisms leading to cancer development [8,71,72]. For instance, the production of acetaldehyde by *Candida* species, especially *C. albicans*, is mainly found in the oral cavity. Acetaldehyde is produced catabolically from its substrate ethanol/glucose by the action of alcohol dehydrogenases (ALDHs) and is found to be elevated in oral carcinomas compared to healthy individuals [55]. The fundamental characteristic features of acetaldehyde are genotoxicity, electrophilicity, effects on DNA repair, and induction of oxidative stress, thereby leading to the formation of protein and DNA adducts and gene mutations. The acetaldehyde-induced formation of DNA adducts interferes with DNA replication, thereby causing point mutations and chromosomal aberrations [73]. Synergistically, they also affect DNA repair and cytosine methylation, stimulating the activation of proto-oncogene and disturbances in the cell cycle, thereby resulting in tumor development. Glutathione, a tripeptide containing glutamate, cysteine, and glycine, functions as a potent anti-oxidant in the antioxidative system [74]. ROS produced during the aerobic metabolism via mitochondrial respiratory chain are well known to be elevated in cancer cells to back their rapid progression [75]. In human neuroblastoma, SH-SY5Y-cell acetaldehyde binds to the glutathione and thereby raises the intracellular ROS and calcium to arbitrate mitochondrial dysfunction [76]. Interestingly, apart from *C. albicans,* non-albicans such as *C. tropicalis* and *C. parapsilosis* are also able to produce significant amounts of acetaldehyde comparable to the carcinogenic levels (>100 μM), which is indicative of the carcinogenic nature of these organisms [77]. 

Glucose metabolism plays a crucial role in the regulation of T-cell activation and cytokine production. Conversion of excessive glucose to acetaldehyde is known to inhibit T cells by downregulating glucose-transporter-1 (*Glut1*) mRNA expression [78,79,80]. For example, exposure of cultured T cells to acetaldehyde (200 μM) reduces glucose uptake with a concomitant reduction in the expression of the *Glut1* transcript. This process also involves the downregulation of hypoxia-inducible factor-1⍺ (*Hif-1**⍺*) mRNA and subsequent suppression of downstream pathways including mammalian target of rapamycin (mTOR), Protein kinase B (PKB or AKT), and translation-initiation factor 4E-binding protein 1 (4E-BP1), all of which are involved in T-cell aerobic glycolysis [80]. Moreover, HIF-1⍺ is known to regulate FOXP3^+^-regulatory-T-cell (T_reg_) differentiation and accumulation [81,82]. Thus, it is clear that alterations in glucose metabolism by *Candida* can impact T-cell function including the suppressive activity by T_regs_. In aging oral mucosa, immune dysfunction is associated with an augmented accumulation of dysfunctional T_regs_ during *Candida* infection (Figure 1) [83]. Similarly, the increased accrual of T_regs_ was also observed in human oral cancers as well in a mouse model of oral cancer [48]. Interestingly, the accumulation of dysfunctional T_regs_ observed in the oral tumor microenvironment somehow positively correlates with the *Candida* burden [48]. However, the role of acetaldehyde expression by *Candida*, metabolic changes in T cells, and their impact on T_reg_ accumulation remains to be seen. Interestingly, Dectin-1, an intermediate in anti-fungal signaling, promotes carcinogenesis. Dectin-1-deficient mice exhibit lowered IL-1β, decreased infiltration of myeloid-derived suppressor cells (MDSC) in the mouse tongues as well as slower progression and a significantly reduced tumor burden. These outcomes significantly correlate with the lower percentage of T_regs_ in the oral tumor environment in these mice [48]. Nevertheless, in the context of tumors, how advanced aging influences *Candida* colonization and infection and alters Dectin-1 signaling remains an intriguing question to be addressed by future studies.

## 3. Role of *Candida* in IBD

Gastrointestinal diseases, predominantly IBD, have emerged as an important public health challenge by globally affecting over 1.5 million annually in North America [84,85]. The causes of IBD remain obscure, though studies on mice and clinical data demonstrate several factors including gut micro/mycobiota, environmental factors, the host’s genetic makeup, etc. that are known to promote disease susceptibility. Microbiota including bacteria, viruses, protozoa, and fungi colonize the mammalian intestine. An intact equilibrium between the resident microbial communities and homeostatic immune responses is indispensable for mammalian health. Recent discoveries have revealed the roles of the fungal community, i.e., mycobiota, in regulating gut immunity as well as the inflammatory origin of human gastrointestinal diseases in their development and progression [86,87].

Several gastrointestinal diseases are interconnected with fungal dysbiosis, which has been shown to contribute to disease sensitivity and severity [88,89]. Antibiotic exposure, genetic diversity, and diet are the underlying factors that promote fungal dysbiosis [87,90,91]. Fungal genera, mostly belonging to the *Candida* and *Saccharomyces* species have been shown to correlate with chronic idiopathic disorders such as ulcerative colitis (UC) and Crohn’s disease (CD), which together are known as IBD [87]. *Candida* species overgrowth in the gut due to antibacterial exposure has been known for a long time [86,87,91]. Antibiotic treatment in the context of oral fungal infection induces not only oral but also gut inflammation [88,89]. 

Recently, the fungal mycobiome has gained an expanded role in promoting human health and disease. Numerous studies have shed light on understanding the role of fungal dysbiosis and altered fungal immune responses in IBD [92,93,94]. In fact, early diagnosis of IBD relies on the prevalence of the anti-*Saccharomyces cerevisiae* antibody (ASCA) among the patients with confirmed IBD, indicating the clinical relevance of yeast in IBD. Explicitly, *C. albicans* and *C. parapsilosis* were consistently found in several IBD cohort studies. Conversely, *Saccharomyces* including *S. cerevisiae* have been found to be reduced in the feces of IBD patients with active inflammation [95,96,97].

The gut fungal composition in patients with CD, UC, and healthy subjects has been evaluated by Liguori et al., who showed that *C. glabrata* predominates during the disease state [97]. Additionally, non-candida fungi such as *Filobasicium uniguttulatum* and *S. cerevisiae* are found to be significantly elevated in the non-inflamed mucosa of CD. On the contrary, UC patients exhibit comparatively lowered fungal diversity [97,98,99]. This could be attributed to the inhibition of antimicrobial peptides against the bacteria in the small bowel modifying the bile-acid reabsorption, thereby benefiting the growth of fungi in patients with CD but not UC. These data also suggest that an increased fungal load of *Candida* species along with altered bacterial diversity could be associated with the pathogenic features of CD. In line with these findings, the 18S rRNA sequence analysis of a colonic biopsy of mucosa tissues and stool samples from patients with UC and healthy individuals revealed marked differences in the fungal communities. UC patients that were profoundly colonized with *C. albicans* displayed heightened mucosal injury and generation of ASCA [100,101,102]. Studies on dextran sodium sulfate (DSS)-induced colitis in C57BL/6J mice usually exhibited fungal invasion to colonic mucosa along with the expansion of *Candida* and *Trichosporon* and a decrease in non-pathogenic *Saccharomyces* spp., [94,102,103]. Numerous studies have shown a high abundance of colonization of *Candida* in ulcer and IBD patients, signifying the association of fungal dysbiosis with IBD [92,96,97,103]. A recent study by Sokol et al. involving the internal transcribed spacer2 (ITS2)-sequencing of rDNA also showed an increase in *C. albicans* and a decrease in *S. cerevisiae* abundance in IBD patients [92].

The relationship between intestinal inflammation and mycobiome dysbiosis appears to be a vicious cycle. Animal studies have also been employed to elucidate the relationship between IBD and the mycobiome. Studies involving the subcutaneous administration of cysteamine, an amino thiol to induce duodenal ulcers have revealed that rats receiving cysteamine and *C. albicans* inoculum progressed with perforated duodenal ulcers in contrast to the rats receiving cysteamine alone [104]. Seminal studies by Iliev et al. showed the protective role of Dectin-1 in colitis in a DSS-induced-colitis mouse model [94]. The study described the association of Dectin-1, which is encoded by the gene C-type lectin domain family 7 member A (*CLEC7A*), with the severity of UC. Mice with Dectin-1-deficiency (*Clec7a*
^-/-^) displayed augmented severity of IBD symptoms, namely worse mucosal erosion, destruction of crypts, and inflammatory cell infiltration than wild-type mice. *Clec7a*
^-/-^ mice also showed an increased production of inflammatory cytokine TNF-α in the colon and an amplified production of IFN-γ and IL-17 in the intestines compared to wild-type littermate controls. The histological examination of colons from DSS-induced *Clec7a*
^-/-^ mice revealed the invasion of fungi in the inflamed tissues while they were localized to the lumens of the wild-type mice. This is in agreement with in vitro studies that showed intestinally conditioned *Clec7a*
^-/-^ DC were less effective in clearing *C. tropicalis*. The study also probed the role of *C. tropicalis* in colitis by comparing it with the non-pathogenic fungi *S. fibuligeria*, where both the yeasts grew in filamentous form and were sensed by Dectin-1. *Clec7a*
^-/-^ mice treated with *C. tropicalis* exhibited augmented colitis symptoms, IFN-γ, and IL-17 produced by T cells from mesenteric lymph nodes and colons compared with non-treated *Clec7a*
^-/-^ controls [86,94,102]. A recent study using Dectin-3-deficient (*Clec4d*
^-/-^) mice showed their susceptibility to DSS-induced colitis. *Clec4d*
^-/-^ bone-marrow-derived macrophages were defective in NF-κB activation induced by *C. tropicalis*, thereby hindering their involvement in tissue repair during fungal invasion [26,105]. Taken together, fungal dysbiosis, impaired anti-fungal immunity and excessive overgrowth of *C. albicans* in the intestine escalate the IBD severity and hinder the gut’s wound-healing processes. 

## 4. *Candida* and Other Fungal Infections in COVID-19

Severe acute respiratory syndrome coronavirus 2 (SARS-CoV-2) is a highly transmissible and pathogenic virus that emerged in late 2019. It has caused a global pandemic of coronavirus disease 2019 (COVID-19), which is an acute respiratory disease threatening human health and public safety [106,107]. As of 24 January 2022, ~349 million confirmed cases and ~5.59 million deaths have been reported globally [108]. SARS-CoV-2 spreads predominantly via respiratory droplets through close person-to-person contact, pre-symptomatic, asymptomatic, and symptomatic carriers. Infections vary from asymptomatic to a full-blown disease with a range of symptoms including upper-respiratory-tract infection or life-threatening sepsis with prolonged hospital stays [106,107,109]. COVID-19 patients show changes in their microbiota composition [110], due to secondary infections including those caused by bacteria and fungi [111]. Among the opportunistic infections in COVID-19 patients, a higher incidence of fungal co-infections was observed [112,113,114,115]. Most fungal-related co-infections in COVID-19 patients involved *Aspergillus* spp., *C. albicans*, *C. auris*, *C. dubliniensis*, *C. glabrata*, *C. tropicalis*, *C. krusei*, and *C. parapsilosis sensu stricto* [116,117]. 

While COVID vaccination has tremendously improved the morbidity and mortality of the disease, secondary fungal infections were common and had worsened the morbidity during the full-blown pandemic until 2021 [114,118]. SARS-CoV-2 triggers host immune dysregulation, which has been postulated to contribute to fungal co-infections increasing the morbidity and mortality associated with COVID-19 [119]. Candidiasis is one of the major complications of severe COVID-19, concomitant with a longer stay in the intensive care unit, a greater usage of broad-spectrum of antibiotics, and a higher mortality rate [115,120,121]. A study by Nucci et al. showed that COVID-19 patients with candidemia were more likely to be under mechanical ventilation compared to non-COVID-19 patients [122,123]. Moreover, *Aspergillus* spp., causes a life-threatening disease-invasive pulmonary aspergillosis (IPA) that is characteristically associated with immune-system dysregulation and affects the respiratory system. IPA chiefly arises due to the prolonged usage of glucocorticoids, immunosuppressive agents, chronic neutropenia, and in transplant recipients [124,125]. Case reports and observational studies have shown that COVID-19-associated pulmonary aspergillosis (CAPA) is of significant concern in hospitalized COVID-19 patients with complications [125,126]. 

Case reports show that patients treated with steroids during COVID-19 treatment often developed histoplasmosis caused by a dimorphic fungus *Histoplasma capsulatum* [127]. Fungus histoplasmosis is highly endemic and has been found in regions of the United States (California), Rio Grande, Southern Brazil, Buenos Aires, and Argentina [124]. This fungus is found in soil containing the excreta of birds or bats [127] and is associated with immunosuppressive conditions [127,128]. Histoplasmosis was previously shown to be associated with acquired immunodeficiency syndrome (AIDS) [129,130], another immune-dysfunction disease. It is noteworthy that this re-emerged as a co-infection in COVID-19 patients [131,132,133]. Another severe opportunistic infection in immune-compromised patients is *Cryptococcus neoformans*, which has a high risk of mortality. Studies have also shown that most COVID-19 patients with *C. neoformans* co-infection did not survive despite treatment with anti-fungal agents such as fluconazole and amphotericin B [114,134]. 

During or after weeks or months of recovery from COVID-19, patients who suffered lung complications have reported experiencing Mucormycosis, commonly known as COVID19-associated Mucormycosis (CAM). CAM was reported in many countries during the COVID-19 second wave including Brazil, Egypt, Austria, Iran, Italy, the United States, France, and India, constituting about 0.3% of COVID-19 infections [135,136]. Mucormycosis is characterized by the presence of hyphal infiltration of the sinus tissue and has a temporal course of roughly four weeks. The common species responsible for Mucormycosis belong to *Rhizopus* spp., *Mucor* spp., and *Cunninghamella* spp. Typical symptoms include complicated sinusitis, crusting nasal blockage, proptosis, facial pain, edema, ptosis, ophthalmoplegia with intracranial extension, and characteristically, a black eschar on the hard palate or in the nasal cavity [114]. For a short period during pandemic, an intensive increase in cases of CAM was observed in several patients requiring extreme surgical operations to treat the disease [114,137,138]. Other fungal co-infections were also described including those by *Pneumocystis jirovecii* and *Coccidioides immitis* [139,140,141]. Some of these co-infections, especially with *P. jirovecii*, are thought to be life threatening or have an adverse impact on COVID-19 [139,141]. 

The concurrent usage of steroids to control COVID-19 symptoms might contribute to the suppression of immunity, and consequently lead to these infections during different stages of the disease, but this possibility remains to be validated. Because steroids may promote these conditions [70,142], a sensible use of these drugs is critical while evaluating the treatment options for COVID-19 and its related co-infections. Additionally, a better understanding of the underlying COVID-19 immune dysfunction is imperative for the appropriate diagnosis and effective treatment of fungal co-infections.

## 5. Established Immune Mechanisms in *Candida* Associated Co-Morbidities

*C. albicans*, the most common member of the intestinal mycobiota, is known to be associated with numerous inflammatory conditions in humans [143,144]. *Candida* is known to poorly interact with epithelial cells. Conversely, its hyphae can enter the epithelial cells and induce numerous pro-inflammatory cytokines and host-defense peptides. The hyphal transition also reduces the colonization potential of *Candida*, typically via the anti-fungal immunity exerted by the augmented expression of hyphal-specific genes [86,145]. Notably, the formation of hyphae in *C. albicans* occurs by its adhesion to the epithelial cells, which is a strong inducer and complemented by hyphal-associated protein expression. Hwp1 (hyphal wall protein 1) and Als3 (agglutinin-like sequence 3) are the two essential proteins that stimulate epithelial adhesion [146,147,148]. These proteins are well-known to possess critical roles in adhesion, invasion, induction of damage, and immune activation/evasion. 

Candidalysin, which is encoded solely by *C. albicans*, is a pore-forming peptide and an essential virulent factor associated with epithelial damage and host immune-cell activation [149]. This peptide stimulates the MPK1/MAPK/cFos-activation pathway, thereby leading to pro-inflammatory cytokine production such as IL-1α, IL-1β, IL-6, IL-8, GM-CSF, and G-CSF. Candidalysin by itself also induces in vitro cell damage in Caco-2 cells; however, ascertaining its effects on the gut epithelium needs additional studies [149,150,151]. Remarkably, candidalysin-mediated epithelial activation does not involve Toll-like receptors (TLRs) or CLRs, demonstrating that these cells utilize sensing mechanisms that are distinct from myeloid cells. This notion is consistent with previous studies that have shown that myeloid cells specifically sense *Candida* cell-wall moieties by β-glucans and mannans while epithelial-cell damage is mediated by the MPK1/MAPK/cFos pathway [147,152,153].

Innate activation through Dectin-1/CARD9 facilitating IL-1β, IL-6, and IL-23 secretion by dendritic cells is critical for disseminating adaptive antifungal Th17-cell responses [31]. A deficiency of Dectin-1 increases the risk of infectious diseases and thus reflects its pivotal role in immune defense [154]. In mice, Dectin-1 deficiency results in a significantly increased fungal burden and a low survival rate [155]. This is associated with a decrease in phagocytic cell recruitment and reduced pro-inflammatory-cytokine release at the site of infection in Dectin-1-knockout mice [155]. Oral candidiasis also showed augmented dissemination in Dectin-1-knockout mice [156]. A deficiency of CARD9, an important intermediate in the Dectin-1/Syk pathway, also results in augmented sensitivity to systemic candidiasis, further reiterating the significance of Dectin-1-mediated signaling in controlling inflammatory diseases involving fungi [28].

mTOR, a conserved Ser/Thr kinase, plays an essential role in the regulation of cell growth and metabolism. mTOR senses and combines several environmental cues, ultimately delivering them to the PI3K-AKT pathway, thereby activating two multiprotein complexes, namely complex1 (mTORC1) and complex2 (mTORC2) [157]. These complexes play divergent roles in T cells, especially in instructing the differentiation of Th cells. By modulating the autophagy/apoptosis pathway, the survival of CD4^+^ T cells during fungal sepsis, as well as CD8^+^-T-cell differentiation, the mTOR pathway regulates the aspergillosis prognosis in mice [158]. Compared with wild-type mice, T-cell-specific mTOR-knockout mice showed a higher survival rate after fungal sepsis, whereas after the T-cell-specific knockout of tuberous sclerosis complex 1(TSC1), which is a negative regulator of mTORC1, mice displayed lower survival rates after fungal sepsis [158,159]. These results show that mTOR activation in CD4^+^ T cells may have a detrimental role in exacerbating inflammation during fungal sepsis. Additionally, mTOR activation impairs CD8^+^-T-cell immunity via the augmentation of Eomesodermin in the context of invasive candidiasis [160]. Moreover, mTOR complexes possess dual and opposing roles in T_regs_, either promoting or inhibiting inflammation depending on the context [161,162,163]. mTOR activation downstream of IL-1β and the myeloid-differentiation-primary-response-88 (MyD88) signaling are critical for T_reg_ homeostasis in oral mucosa. Consequently, the mTOR/IL-1β/MyD88-signaling axis promotes the induction and expansion of an exclusive population of FOXP3^+^ cells expressing IL-17A, known as T_reg_17 [83]. However, the dysregulation of any node in this axis appears to diminish these cells and cause increased accumulation of dysfunctional T_regs_. Additionally, it appears that aging and tumorigenesis may disrupt this signaling and potentially cause *Candida* dysbiosis and infection. Thus, mTOR appears to regulate immune dysfunction, dysbiosis, and *Candida*-associated oral inflammation [83]. The role of this signaling in gut dysbiosis or in relevance to other diseases remains unclear and warrants further investigation. Taken together, emerging roles of fungi in altering various mechanistic components including mTOR highlight their critical functions in skewing the regulation of T-cell metabolism/differentiation and mucosal homeostasis. 

## 6. Concluding Remarks

Fungal dysbiosis and altered immune responses to fungi are linked to several human pathologies. Most fungal infections represent polymicrobial diseases in susceptible hosts and are more prevalent in immune-compromised and elderly individuals. *Candida* infections have steadily increased because of a higher incidence of systemic diseases including tumors, IBD, AIDS, disorders of the kidney and liver, the extensive development of interventional therapy, transplantations, and the over-exploitation of various antibiotics. Lymphocytopenia is commonly observed in immunocompromised/elderly population/COVID-19 patients, making them prone to persistent fungal infections or non-resolving inflammation associated with such infections. Over the last decade, a tremendous advancement has been made in understanding anti-fungal immunity. However, further understanding of the precise mechanistic interplay between fungal dysbiosis and mucosal immune system and its impact on the host in the context of diseases needs additional detailed investigations. Those studies will lead to potential new targets for reversing immune dysfunction in the context of fungal dysbiosis.

## Figures and Tables

**Figure 1 pathogens-11-00212-f001:**
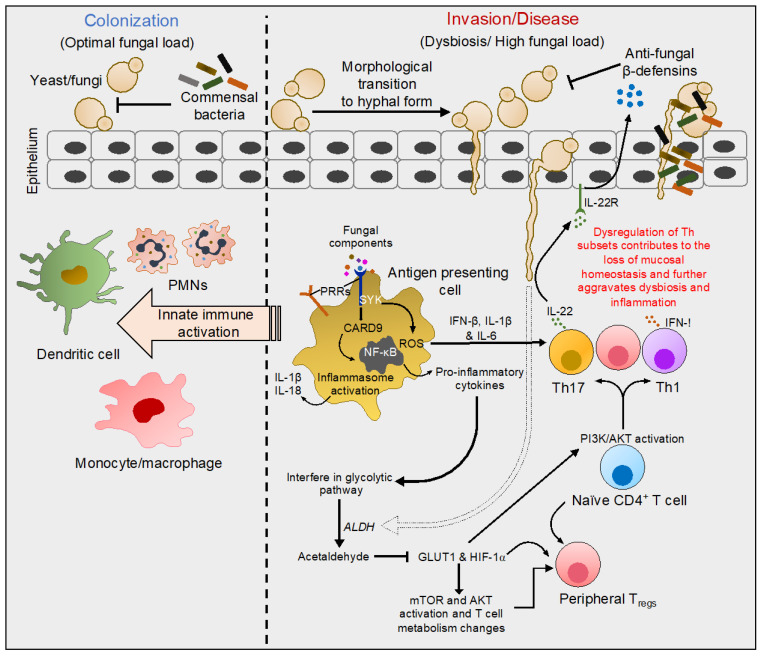
Molecular events characteristic of interactions of fungi with mucosal surfaces. Fungi including *C. albicans* colonize healthy human skin and other mucosal surfaces in yeast form. In healthy individuals, yeast colonization does not induce epithelial damage or an intense cytokine response. During infection or dysbiosis in the context of diseases, yeast transform into hyphal form, gaining access to underlying host tissues (albeit hyphal transition does not occur in certain species including *C. glabrata*). Fungal components such as mannans, mannoproteins, β-glucans, and chitin are sensed by innate antigen-presenting cells through pattern-recognition receptors (PRRs) on their surface. Dectin-1 activation and the phosphorylation of the receptor at the cytoplasmic domain leads to the activation of spleen tyrosine kinase (Syk). Upon activation, a series of events leads to the assembly of a complex that activates the NF-kB pathway. NF-kB activation results in the release of pro-inflammatory cytokines and reactive oxygen species (ROS) triggering phagocytosis. These events also lead to inflammasome activation, triggering IL-1β and IL-18 production that further prime the innate immune activation. Some of the cytokines modulate PI3K/AKT and glycolytic pathways, instructing T-helper (Th)1, Th17, and regulatory T-cell (T_reg_) polarization and their functions. Th17 cells additionally promote the production of innate anti-microbial proteins. For example, pro-inflammatory cytokines secreted by macrophages induce a Th17 response releasing IL-22, which stimulates defensin secretion by epithelial cells. Besides the activation of the immune system, direct production of acetaldehyde resulting from *Candida* metabolism inhibits the expression of glucose transporter1 (GLUT1) and hypoxia-inducible factor-1α (HIF-1α), which subsequently may also control Th and T_reg_ homeostasis. Thus, fungal dysbiosis modulates local immune milieu and alters Th functions, which significantly switches the balance between health and disease states.

## Data Availability

Not applicable.

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
