# Peer review of "Fungal Colonization and Infections—Interactions with Other Human Diseases"

_pathogens, 2022, doi:10.3390/pathogens11020212_

Round 1
Reviewer 1 Report
General Impression
The authors present a broad review of selected aspects of fungal contributions to common diseases. The manuscript shows that team has sufficient expertise and good insights into fungus-host interactions, which allowed them to write an insightful paper summarizing past and future directions of research. References are current and are cited appropriately, and - to the best of this reviewer’s knowledge – the conclusions are accurate. There are, however, some issue with the organization and writing of the manuscript that need to be addressed. Since this a review article that does not present original research, no further controls or other experiments are necessary.
Scope of review
The article summarizes the current state of knowledge on host-fungal interactions in cancer, inflammatory bowel diseases and COVID. It is not clear why these 3 areas have been chosen, and to which audience the manuscript is directed. In order to avoid the impression that the selection has been made based on convenience, it would be helpful to stress out common themes of interactions in these three very different pathologies. A more integrated approach to writing would have avoided duplications, e.g. of the mechanism of dectin signaling that is discussed in lines 61 and 400.
Style and grammar
The manuscript needs a thorough review of style, grammar and semantics. There are numerous examples of incomplete sentences that obscure the meaning of the statement (e.g.line 402: “In vitro studies show following epithelial disruption;”), word duplications that impact readability (e.g. 432: “Also, a recent study shows mTOR has been shown to regulate…” and odd choices of words that describe a finding inaccurately. Consider as an example line 171: “Furthermore, overexpression of Ki-67, P53 and COX-2 by Candida can also affects the malignant transformation into oral leukoplakia.”. First, Candida does not overexpress these factors – rather, host cells overexpress them in response to Candida infection. Second, there is a subject verb-disagreement in the sentence. And third, the referenced article (66) does not discuss the expression of the tumor suppressor P53 – it shows instead the role of the tumor promoter P16.
Reviewer 2 Report
The legend for Figure 1, should state that germ tube formation/hyphal development may occur with some yeasts. There are other yeasts in the GI tract beside Candida; for instance, C. glabrata, does not make hyphae but certainly gets into the intravascular compartment.
Reviewer 3 Report
I really enjoyed this paper immensely it was beautifully and carefully written making it a joy to read. This paper is going to be a staple for what I have new students read when they join my laboratory. That being said I believe that there are ways to make this review even better and an easier read for the readers.
Here are a few of my comments:
- for line 122 you title it Candida in Cancer. I would suggest that you change to better fit what you are reviewing to help the readers understand better. In lines 130-133 you seem to indicate that you are not going to discuss how Candida affects cancers. Yet given your 3 points and the line 134 it would that you are going to talk about candida aiding in cancer and/or causing it. This is confusing for the reader. A clearer introduction as to precisely what will be reviewed would help the readability and reduce confusion.
- for the paragraph line 184 I would suggest a figure for clarification as fig 1 did previously
- For the paragraphs from 184 through 227 The material is explained in a very dense manner as stated in 2. I would suggest that you try to expand your explanations and use a figure to better guide the readers. This section is the weakest section of the paper and I believe expanding or describing in more detail will bring it up to the same level as the rest of the paper.
- Sentence 449 is confusing
- line 450 you have "Fungal dysbiosis and altered immune responses to fungal". I think you mean fungi
Round 2
Reviewer 1 Report
The revision has resulted in a much improved manuscript and I have no further suggestions for improvement.